# Empowering medical students: Peer-Led OSCE reduces anxiety and may enhance test performance

**Leonardo Mateus de Lima**[1,2*], **Maria Helena Favarato**[1,2],
**Iolanda Fátima Lopes Calvo Tibério**[1]

**1** Department of Internal Medicine, Medical School, University of São Paulo, São Paulo, Brazil, **2** Health School, Municipal University Of São Caetano do Sul, São Paulo, Brazil

* leonardo.mlima@fm.usp.br

## Abstract

### Background

The Objective Structured Clinical Examination (OSCE) is a widely used assessment method often associated with high levels of student anxiety and significant logistical demands, including time and professional resources. This study evaluated the impact of a peer-led mock OSCE intervention—wherein students conceptualized and administered the mock assessment—on anxiety levels and subsequent performance in summative OSCE.

### Methods

Intervention: At the end of 2023, 28 third-year medical students taking the summative OSCE for the first time were randomized to participate in a peer-led mock OSCE, taking on roles as station designers, examiners, and examinees. An additional 39 students formed the control group, who did not participate in the intervention. Endpoints: Primary endpoint was state anxiety levels assessed with the State-Trait Anxiety Inventory (STAI); summative OSCE performance scores were a secondary endpoint.

### Results

Students in the mock OSCE group demonstrated significantly lower state anxiety levels ($54.80 \pm 13.40$) than the control group ($63.56 \pm 12.20$, $p = 0.009$, Cohen's $d = 0.692$, mean difference $= 8.79$ points, 95% CI [2.25, 15.3]). Summative OSCE scores were numerically higher in the intervention group ($8.60 \pm 0.63$) compared to control ($8.30 \pm 0.85$, $p = 0.055$, $r = 0.271$, 95% CI [−0.000019, 0.600]), showing a favorable directional trend.

**Data availability statement:** All relevant data are within the manuscript and its Supporting Information files.

**Funding:** This study was financially supported by the Fundação de Amparo à Pesquisa do Estado de São Paulo (FAPESP) through the Thematic Project (Grant nº 2018/02537-05 to IT).The funders had no role in study design, data collection and analysis, decision to publish, or preparation of the manuscript.

**Competing interests:** The authors have declared that no competing interests exist.

## Conclusion

Participation in a peer-led mock OSCE significantly reduced state anxiety levels and may improve summative OSCE scores. This intervention may represent a valuable strategy to enhance both academic outcomes and student well-being in high-stakes medical assessments.

---

## Introduction

The Objective Structured Clinical Examination (OSCE), introduced by Harden et al. in 1975 [1], has become one of the most effective methods for assessing clinical skills in medicine [2], residency programs, and other health science fields such as nursing, pharmacy, and physiotherapy [3–4].

While OSCE is often used for summative assessments, it is also a powerful tool for formative purposes [5–6], providing immediate feedback [7]. However, its use in formative assessments is often limited by the high costs associated with its implementation [8–9]. Administering an OSCE requires significant time and personnel, including both examiners and simulated patients [10]. These logistical and financial constraints hinder the routine use of OSCE as a formative tool throughout the semesters.

An alternative to these limitations is engaging volunteer students as simulated patients and even as examiners [11]. Young et al. described an initiative in which senior medical students organized an OSCE for their junior peers, an approach that was well received by participants [12]. Since then, many other authors have published similar experiences.

Another widely recognized challenge of OSCE is its potential to induce stress and anxiety in students [13]. To address this, various strategies have been proposed [14–15], including student-led initiatives in organizing mock OSCEs, which have shown promise in reducing anxiety and improving preparedness [16–17]. However, this strategy does not appear to improve performance in exams [18].

Despite numerous studies involving students in the organization of both summative and mock OSCEs, few have specifically evaluated students from the same cohort, focusing on outcomes such as performance and anxiety levels.

Therefore, this study aimed to investigate the impact of a peer-led mock OSCE on students' anxiety and their performance in the summative OSCE.

## Methods

### Ethical approval

Prior to volunteer recruitment, the Municipal University of São Caetano do Sul ethics committee approved the study (CAAE: 63711722.6.0000.5510). Every student gave written consent to their participation, data collection and anonymous publication of the results.

### OSCE

In Brazil, medical school lasts six years, and many universities have incorporated the OSCE as a clinical skills assessment tool in both clinical and pre-clinical courses. At

the Municipal University of São Caetano do Sul, medical students take the OSCE for the first time at the end of the fifth semester in a course called Medical Skills. The exam consists of five stations, each lasting six minutes and comprising one to three tasks. Each station has a professor as an examiner and, depending on the scenario, may include a simulated patient or a mannequin for procedural tasks.

Students receive a list of ten possible themes four weeks before the summative OSCE. All themes are covered during the semester through lectures, problem-based learning (PBL) activities, and realistic simulation or activities with standardized patients. PBL and simulation sessions are facilitated by experienced tutors responsible for the creation, organization, and administration of the OSCE, under the supervision of the Medical Skills course coordinator.

## Study design

A prospective, randomized, controlled, and non-blinded study was conducted to evaluate whether participation in a student-led mock OSCE reduces anxiety and enhances performance in the summative OSCE. Two study groups were formed after randomization: intervention and control.

All 5th-semester medical students were invited to participate via institutional email and a messaging platform. The recruitment period lasted from October 16 to November 16, 2023. After compiling the final list of volunteers, simple randomization was performed using Microsoft Excel (version 2507). Each student was assigned a random number, and participants were then ordered accordingly and allocated alternately into the intervention or control group. No stratification was used to balance demographic or academic characteristics.

A total of 35 students were assigned to the intervention group, with 15 designated as station creators and examiners, and 20 as examinees. The remaining students were allocated to the control group.

## Endpoints

Anxiety levels, measured using the State-Trait Anxiety Inventory (STAI) state subscale, were defined as the primary endpoint of the study.

The secondary endpoint was performance on the summative OSCE, assessed by the final exam score.

## Intervention

The intervention group was divided into two subgroups: examiners and examinees. One week before the summative OSCE, a preliminary meeting was held with the examiners' subgroup to develop five OSCE stations under the supervision of a professor. The supervising faculty member served to provide foundational exam guidance, allocate station topics randomly, and rectify content-specific inaccuracies. However, the professor did not modify the station structure—only inaccuracies in medical knowledge. Participants were free to use their own knowledge, books, journals, and online resources to design their stations.

A few days later, the examiners' subgroup assessed the examinees' subgroup in the mock OSCE. The control group consisted of students randomized to this group and those unable to participate in the mock OSCE due to academic scheduling conflicts. Finally, all students took part in the summative OSCE.

An online course was provided to all students and professors, covering key aspects of OSCE, including historical background, clinical skills assessment methods, station and checklist design, and exam organization.

## Summative OSCE and data collection

One week after the intervention, the summative OSCE was conducted. It consisted of five stations, each lasting six minutes, covering the topics described in Table 1. Each station had a checklist scored from 0 to 10, completed by a faculty member. None of the faculty members involved in the summative OSCE had participated in or had prior knowledge of the stations from the simulated OSCE. The final OSCE score was calculated as the arithmetic mean of all station scores.

**Table 1. Themes and clinical skills assessed in the mock and summative OSCE.**

| OSCE | Theme | Skills |
|------|-------|--------|
| *Summative* | | |
| Station 1 | Abdominal pain | History taking, physical examination, clinical reasoning |
| Station 2 | Low back pain | History taking, physical examination |
| Station 3 | Ascites | History taking, physical examination |
| Station 4 | Pleural Effusion | Physical examination, imaging interpretation |
| Station 5 | LGBTQIA+ health | History taking, communication |
| *Mock* | | |
| Station 1 | Low back pain | History taking, physical examination |
| Station 2 | Heart Failure | History taking, physical examination |
| Station 3 | Trigeminal Neuralgia | History taking, physical examination |
| Station 4 | Pleural Effusion | Physical examination, communication |
| Station 5 | LGBTQIA+ health | History taking, communication |

OSCE = Objective Structured Clinical Exam. LGBTQIA+ = lesbian, gay, bisexual, transgender, queer/questioning, intersex, asexual, and other identities not included in the acronym.

During the official OSCE, all students included in the study completed the state anxiety questionnaire from the STAI immediately after finishing the last station.

The State-Trait Anxiety Inventory, developed by Spielberger in 1970 [19], is a validated tool for assessing anxiety and consists of two subscales: state anxiety (STAI-S) and trait anxiety (STAI-T), each comprising 20 items. State anxiety is a transient emotional response directly related to a specific stressor, such as an examination [20]. In this study, only the state anxiety subscale was used. Each item is scored on a four-point Likert scale ("Not at all," "Sometimes," "Usually," and "Very much so"), generating a total score ranging from 20 to 80, with higher scores indicating greater anxiety levels.

Finally, to assess whether potential differences in performance on the summative exam could reflect selection bias, the grades from all other courses taken during the same semester as the intervention were collected, and a global average was calculated, referred to as "academic score" in this study. For this calculation, data from the Medical Skills course were excluded.

## Statistics

For statistical analysis, BioEstat 5.3 and Jamovi 2.6.44 software were used, with a significance level of $p < 0.05$. Normality was assessed using the Shapiro-Wilk and D'Agostino tests. The Student's t-test was used to compare means, while the Mann-Whitney U test was applied for non-parametric data. Correlations were analyzed using Pearson or Spearman tests, depending on data distribution.

The sample size calculation was based on state anxiety as the primary endpoint, considering a 10-point difference in STAI scores as clinically relevant. A pilot study conducted with a previous cohort of 65 fifth-semester students revealed a mean STAI score of 59.8 and a standard deviation of 12.83. Using these values, and assuming a two-tailed test, a significance level (α) of 0.05, and 80% power (1–β), the required sample size was determined to be 27 participants per group. To account for an estimated 10% dropout rate, we increased the target sample size to 30 students per group.

All relevant data are within the paper and its Supporting Information files.

## Results

### Mock OSCE (peer-led)

The participants randomized to design the mock OSCE (n = 12) devoted four hours to developing the exam. First, pairs or small groups (two to three students) were assigned a station theme and developed the scenario and checklist based on

objectives determined by themselves. Once all stations were created, the students reconvened to review and refine the scenarios and checklists. The professor ensured time management and corrected conceptual errors without altering the structure of the stations.

Two days later, the students who designed the mock OSCE acted as examiners for their peers (examinees, n = 16). Each station lasted six minutes, followed by one additional minute for immediate feedback. Table 1 summarizes the station themes, and the clinical skills assessed.

## Population

All 67 fifth-semester students volunteered and were eligible for this study. After randomization, 35 volunteers were assigned to the intervention group. However, 7 students were unable to participate in the mock OSCE due to academic scheduling conflicts and were subsequently incorporated into the control group (Fig 1).

For OSCE performance analysis, data from all students was available. However, for state anxiety analysis, 5 students were excluded due to incomplete STAI form submissions.

Table 2 presents the demographic characteristics of the study population. The mean age was 26.46 years, and females predominated in the cohort (80.6%), comprising 89% of the intervention group and 74% of the control group.

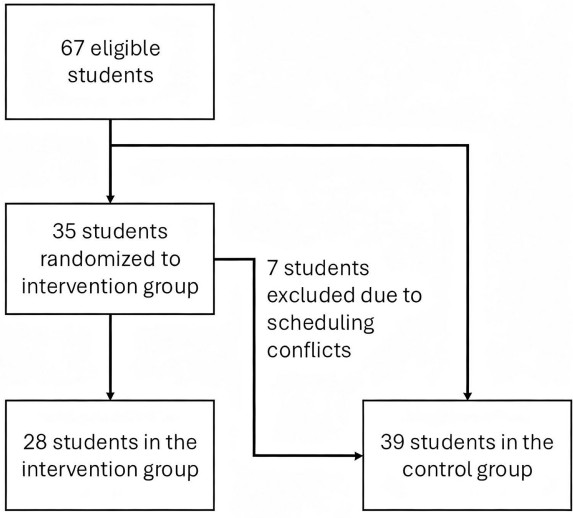

**Fig 1. Flow chart.** Distribution of students after randomization.

**Table 2. Characteristics of study participants.**

| Group | n | Age (years) | gender (%) | academic score (mean±SD) |
|---|---|---|---|---|
| Control | 39 | 23.36 | female, 74% | 7.78 ± 0.63 |
| Intervention | 28 | 23.60 | female, 89% | 7.86 ± 0.56 |
| Examinator | 12 | 24.25 | female, 100% | 7.99 ± 0.60 |
| Examinee | 16 | 23.13 | female, 81% | 7.77 ± 0.52 |

Demographic characteristics of study participants. The control group included 39 students, while the intervention group included 28 students, subdivided into 12 examiners and 16 examinees. Variables presented are age (years), gender (percentage of females), and academic score (mean±SD). SD = standard deviation.

## State anxiety

The primary endpoint of the study was state anxiety. As shown in Fig 2, students in the intervention group had significantly lower state anxiety levels (54.8 ± 13.40, mean ± standard deviation) compared to the control group (63.56 ± 12.20, p = 0.009; mean difference = 8.79, 95% CI [2.25, 15.3], Cohen's d = 0.692). Subgroup analysis further confirmed this difference, with both examiners (55.55 ± 13.49, p = 0.035) and examinees (54.20 ± 13.70, *p* = 0.01) exhibiting lower state anxiety levels than the control group. However, no significant differences were found between examiners and examinees within the intervention group. Additionally, as illustrated in Fig 3, there was no significant correlation between state anxiety and OSCE performance (Spearman's correlation, rs = −0.18, p = 0.17).

## OSCE performance

Regarding student performance in the summative OSCE (Fig 4), the intervention group achieved higher scores (8.60 ± 0.80, median ± IQR) compared to the control group (8.30 ± 0.90). Although the Mann-Whitney U test did not reach conventional statistical significance under a two-tailed assumption (U = 395, p = 0.055), there was a non-significant trend favoring the intervention group. The median difference was 0.30 points (95% CI [−0.000019, 0.600]), and a small effect size was observed (Rosenthal's r = 0.277). Subgroup analysis revealed no differences between examiners and examinees; however, examiners (8.65 ± 0.48) performed significantly better than the control group (*p* = 0.006).

Table 3 provides detailed subgroup analysis and the performance of each group across individual OSCE stations. In Station 1, the examiner subgroup outperformed the control group (9.25 ± 0.63 vs. 9.00 ± 1.25; p = 0.01). For Station 5, the intervention group (10 ± 1.00) scored higher than the control group (9.00 ± 2.00; p = 0.033). No significant differences were observed in Stations 2, 3, and 4. The stations themes and clinical skills assessed are summarized in Table 1.

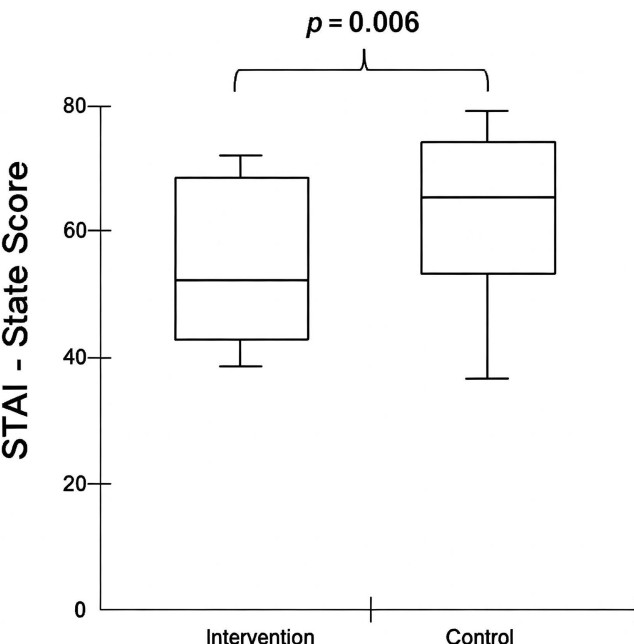

**Fig 2. State anxiety score.** The intervention group had significantly lower state anxiety levels (54.8 ± 13.40, mean ± standard deviation) compared to the control group (63.56 ± 12.20, *p* = 0.009; Cohen's d = 0.692, 95% CI [0.154, 1.19]), indicating a moderate effect size. Student's t test. STAI: state-trait anxiety inventory.

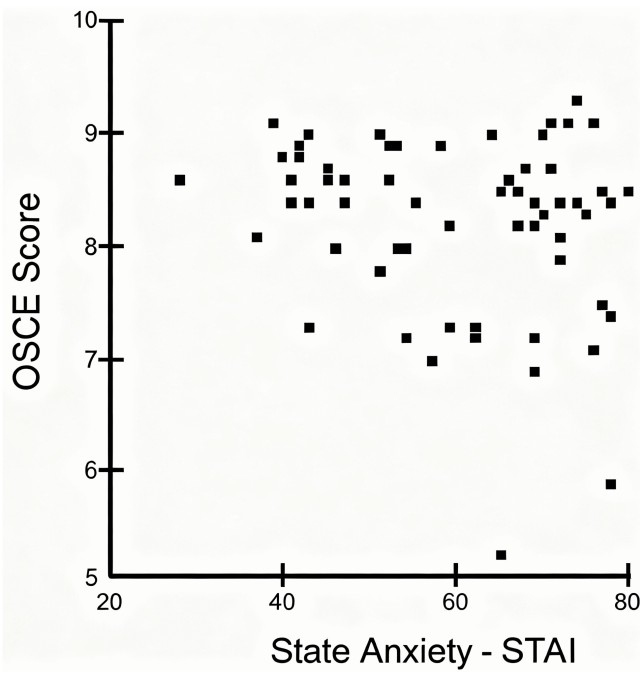

**Fig 3. Correlation between OSCE scores and state anxiety.** Correlation between Objective Structured Clinical Examination (OSCE) scores and State-Trait Anxiety Inventory (STAI) state anxiety scores in medical students (n = 62). A weak, non-significant negative correlation was found between OSCE performance and state anxiety (rs = −0.18; $p$ = 0.17).

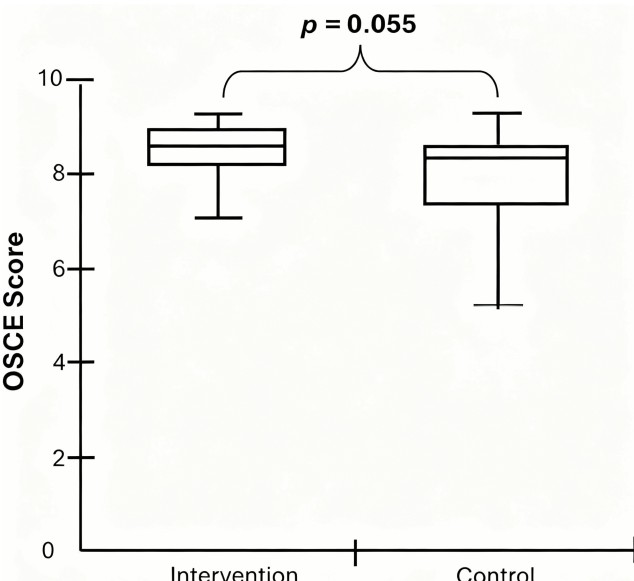

**Fig 4. OSCE Performance.** Objective Structured Clinical Examination (OSCE) scores for the Intervention and Control groups. The intervention group achieved significantly higher OSCE scores (median = 8.60, IQR = 0.80) compared to the control group (median = 8.30, IQR = 0.90; $p$ = 0.055, Mann-Whitney U test), with a small effect size (Rosenthal's r = 0.277; 95% CI [−0.000019, 0.600]). IQR = interquartile range. CI = confidence interval.

**Table 3. Performance of each group and subgroup across OSCE stations.**

| | | Examinator | Examinee | Intervention | Control | p |
|---|---|---|---|---|---|---|
| | | n = 12 | n = 16 | n = 28 | n = 39 | |
| Station 1 | mean ± SD | 9.29 ± 0.54 | 8.59 ± 1.14 | 8.89 ± 0.98 | 8.54 ± 0.98 | *0.01 |
| | median ± iqr | *9.25 ± 0.63 | 8.50 ± 2.5 | 9.00 ± 1.63 | *9.00 ± 1.25 | |
| Station 2 | mean ± SD | 7.63 ± 1.30 | 7.31 ± 1.30 | 7.45 ± 1.23 | 7.36 ± 0.87 | ns |
| | median ± iqr | 7.75 ± 1.25 | 7.50 ± 1.25 | 7.50 ± 1.63 | 7.50 ± 1.50 | |
| Station 3 | mean ± SD | 7.63 ± 1.65 | 7.25 ± 1.51 | 7.41 ± 1.55 | 7.36 ± 0.87 | ns |
| | median ± iqr | 8.00 ± 1.50 | 7.75 ± 1.63 | 8.00 ± 1.75 | 7.50 ± 2.00 | |
| Station 4 | mean ± SD | 9.00 ± 0.74 | 8.53 ± 1.44 | 8.73 ± 1.20 | 7.26 ± 1.91 | ns |
| | median ± iqr | 8.50 ± 1.50 | 8.50 ± 2.00 | 8.50 ± 1.50 | 8.50 ± 2.00 | |
| Station 5 | mean ± SD | 9.50 ± 0.80 | 9.25 ± 1.24 | 9.36 ± 1.06 | 8.77 ± 1.37 | *0.054 † 0.033 |
| | median ± iqr | *10 ± 1.00 | 10 ± 1.00 | † 10 ± 1.00 | *† 9.00 ± 2.00 | |
| OSCE | mean ± SD | 8.61 ± 0.53 | 8.19 ± 0.75 | 8.37 ± 0.60 | 8.06 ± 0.84 | *0.006 † 0.055 |
| | median ± iqr | *8.65 ± 0.48 | 8.35 ± 1.30 | † 8.60 ± 0.80 | *† 8.30 ± 0.90 | |

Performance scores across Objective Structured Clinical Examination (OSCE) stations. Data are presented as mean ± standard deviation and median ± interquartile range for the Examinator, Examinee, Intervention, and Control groups. The Intervention group is the combination of the Examinator and Examinee subgroups. Significant p-values ($p < 0.05$, Mann-Whitney U test) for the comparison between the entire Intervention group and the Control group are indicated in the respective row for the total OSCE score. The symbols denote specific pairwise comparisons: (*) Examinator vs. Control subgroup; (†) Intervention vs. Control group. SD = standard deviation. ns = non-significant.

Finally, this study did not find any difference in the academic score between intervention (7.86 ± 0.55) and control group (7.77 +− 0.63, p = 0.35, Mann-Whitney U Test).

## Discussion

In the present study, participation in a peer-led-mock OSCE was associated with lower anxiety levels during the summative OSCE among fifth-semester medical students. Similar findings have been reported, particularly in studies based on subjective assessments and self-reported measures [12,13,16]. Nonetheless, none of these studies have investigated the effects of students from the same cohort organizing their own mock OSCE on anxiety. Most works proposed faculty-led-mock OSCE or near-peer-led OSCE.

Evidence suggests that participating in a mock OSCE can enhance students' confidence [16], which may contribute to lower anxiety scores. Familiarity with the exam's structure and overall functioning can help mitigate the uncertainty associated with the official assessment. Additionally, the mock OSCE serves as a formative experience, offering valuable feedback that allows students to refine their knowledge and further build their confidence.

Although OSCE is associated with higher levels of anxiety [21–23], it does not necessarily mean that higher anxiety leads to lower performance. Consistent with previous data [24], the study findings showed no significant correlation between anxiety levels and OSCE performance. One possible explanation is that as students progress, their cognitive maturity and ability to tolerate high cognitive load may weaken the association between test anxiety and performance [24].

While the intervention group's performance on the summative OSCE was only modestly higher, the observed trend suggests that peer-led mock OSCEs may contribute to improved preparedness and performance. Although the difference was not statistically significant, the direction and consistency of the data across stations indicate a potentially meaningful educational effect that warrants further investigation. The literature on their efficacy for improving summative exam performance, however, is conflicting. For instance, Madrazo [18] found no improvement in final exam scores after a mock OSCE where students participated solely as examinees. In contrast, Heinke [25] demonstrated objective benefits in summative evaluations when students were engaged in the entire process, including station creation and exam organization.

Heinke [25] argues that the observed results may be attributed to the time students spent engaging with the theoretical and practical content of the medical skills course. This aligns with the so-called *generation effect*, in which individuals better remember materials they have generated themselves [26]. This is consistent with the subgroup analysis, which showed differences in the first OSCE station—a topic not covered in the formative activity. Furthermore, although not statistically significant, there was a trend toward better performance in all stations of the official exam among the experimental groups. Station 5 (LGBTQIA+ health), however, closely resembled one of the stations created in the mock OSCE, which could suggest that content familiarity, in addition to format familiarity, may have contributed to the improved performance on that specific station.

When students design and participate in mock exams, they actively practice the required skills, which can be applied across different scenarios. For instance, practicing history-taking and physical examination for a patient with pneumonia can develop skills that are also useful for a station on pleural effusion. Similarly, communication skills training can be beneficial across multiple scenarios that assess doctor-patient interaction.

When participating as an examiner, the student can identify and reflect on strengths and fragilities of other students during the OSCE stations, which may help improve their own performance during a summative assessment [27].

Student performance in OSCEs is influenced by multiple factors [28], with anxiety representing only one of them. Knowledge acquisition, familiarity with exam structure, self-efficacy [29], and test-taking skills are also key determinants of performance [28]. In this study, no clear correlation was observed between state anxiety and OSCE scores, reinforcing the notion that performance outcomes result from a complex interplay of cognitive, emotional, and contextual variables rather than from anxiety levels alone.

Although not a primary research question in this study, a peer-led mock OSCE appears to be a cost-effective strategy for helping students manage stress and anxiety. Additionally, it may contribute to improved performance in summative skills assessments.

The study lacked sufficient power to confirm whether the intervention improves performance and was unable to determine the specific impact of different roles in a mock OSCE, such as station creation, examiner participation, and being an examinee. Additionally, the control group consisted of students who could not participate in the mock OSCE due to logistical reasons, which may have introduced variability in baseline characteristics.

Selection bias is a potential concern, since participation was voluntary. Nevertheless, all 5th-semester students agreed to participate, and the final sample represents a substantial portion of the cohort. Randomization was performed only after the volunteer list was complete, ensuring balanced group allocation. Seven students initially assigned to the intervention group could not attend the mock OSCE and were reassigned to the control group before the intervention, thereby preserving the integrity of group comparisons.

The generalizability of the study findings should be interpreted with caution. This was a single-center study with a relatively small sample size and a predominance of female participants (over 70%). As such, the results may not be fully representative of broader medical student populations with different demographic or institutional profiles. Future multi-center studies with more diverse and larger samples are needed to confirm these findings.

However, this study has several strengths. First, it was randomized and controlled. Second, to the authors' knowledge, this is the first study to objectively assess the impact of a peer-led mock OSCE on anxiety levels in students from the same cohort, all enrolled in the same semester of medical school. Third, although the study was not fully blinded, OSCE evaluators were unaware of the participants' group assignments, reducing the risk of assessment bias. Finally, engaging students in creating OSCE stations and acting as examiners promotes active learning, which is a key educational strength of the study design.

## Conclusion

This investigation demonstrates that among students undertaking a summative OSCE for the first time, participation in a peer-led mock OSCE is associated with lower anxiety levels and may possibly enhance test performance. This

intervention may represent a valuable strategy to enhance both academic outcomes and student well-being in high-stakes medical assessments.

## Supporting information

**S1 Dataset. Minimal dataset containing anonymized information on student group allocation, state anxiety scores, and OSCE performance used in the analyses presented in this study.**
(XLSX)

## Acknowledgments

The authors are sincerely grateful to the faculty and staff of Municipal University of São Caetano do Sul for their valuable support in conducting this study. They also extend their appreciation to the students, who are the very reason for studies such as this to exist.

## Author contributions

**Conceptualization:** Leonardo Mateus de Lima, Maria Helena Favarato, Iolanda Fátima Lopes Calvo Tibério.

**Data curation:** Leonardo Mateus de Lima.

**Formal analysis:** Leonardo Mateus de Lima.

**Investigation:** Leonardo Mateus de Lima.

**Methodology:** Leonardo Mateus de Lima, Maria Helena Favarato.

**Supervision:** Maria Helena Favarato, Iolanda Fátima Lopes Calvo Tibério.

**Visualization:** Maria Helena Favarato.

**Writing – original draft:** Leonardo Mateus de Lima.

**Writing – review & editing:** Leonardo Mateus de Lima, Maria Helena Favarato, Iolanda Fátima Lopes Calvo Tibério.

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
