## [Decision Letter · Decision Letter 0]

19 Jul 2025

Dear Dr. Mateus de Lima,

Thank you for submitting your manuscript to PLOS ONE. After careful consideration, we feel that it has merit but does not fully meet PLOS ONE’s publication criteria as it currently stands. Therefore, we invite you to submit a revised version of the manuscript that addresses the points raised during the review process.

We look forward to receiving your revised manuscript.

Kind regards,

Lorenzo Faggioni, M.D., Ph.D.

Academic Editor

PLOS ONE

Journal Requirements:

Reviewers' comments:

Reviewer's Responses to Questions

**Comments to the Author**

1. Is the manuscript technically sound, and do the data support the conclusions?

Reviewer #1: Partly

Reviewer #2: Yes

2. Has the statistical analysis been performed appropriately and rigorously?

Reviewer #1: Yes

Reviewer #2: Yes

3. Have the authors made all data underlying the findings in their manuscript fully available?

Reviewer #1: Yes

Reviewer #2: Yes

4. Is the manuscript presented in an intelligible fashion and written in standard English?

Reviewer #1: Yes

Reviewer #2: Yes

Reviewer #1: Manuscript title: Empowering Medical Students: Peer-Led OSCE Reduces Anxiety and Enhances Test Performance

Major Comments:

1. randomization and sample composition: the manuscript would benefit from a more detailed account of the randomization process. It is currently unclear how the allocation sequence was generated or whether stratification was employed to balance the demographic or academic variables across groups. Additionally, the potential for selection bias should be acknowledged, particularly because students volunteered to participate in the mock OSCE. This self-selection could influence both anxiety levels and performance outcomes of the participants.

2. power analysis and effect size: while a power analysis is mentioned, the underlying assumptions are not provided in the manuscript. The authors should specify the expected effect size, standard deviation, and dropout rate used for the calculation. This information is important for assessing whether the sample size was sufficient to detect meaningful differences between groups. Reporting effect sizes for the main comparisons further contextualizes the statistical findings.

3. content overlap between assessments: there is a noted overlap in the content between the mock OSCE stations and the summative exam. This raises the possibility that performance improvements were influenced by familiarity rather than anxiety reduction. Although the authors briefly acknowledge this issue, a more explicit discussion of its potential impact on the results is required. The distinction between cognitive reinforcement and affective modulation must be clearly highlighted.

4. interpretation of performance outcomes: the observed difference in OSCE scores, although statistically significant, was modest. The manuscript should address the clinical or educational relevance of these differences. Would this change have any implications for grading thresholds, remediation decisions or student confidence in clinical environments? Clarifying this would enhance the practical utility of these findings.

Minor Comments

5. The manuscript does not specify whether the assumptions of normality were tested before applying parametric tests. Given the small sample size, this needs to be confirmed. A brief note on how the data distribution was assessed would be enough.

6. writing and redundancy: the text includes several repetitive elements, particularly in the Abstract and Introduction sections. The notion that OSCEs are anxiety-inducing was stated multiple times with minimal variation. Streamlining these sections will improve readability.

7. visual representation of data: figures and tables are informative, but the presentation could be enhanced by including graphical summaries of anxiety scores and OSCE outcomes. Box plots or bar graphs with confidence intervals could facilitate reader interpretation.

Reviewer #2: This is a well-structured and timely study that addresses a relevant challenge in medical education. The idea of a peer-led OSCE is innovative and clearly has practical implications for reducing test anxiety and potentially improving performance. The manuscript is generally clear and the methodology is appropriate. However, I encourage the authors to expand their discussion of potential biases (e.g., self-selection, lack of blinding) and improve statistical reporting by including effect sizes and confidence intervals. Addressing these points will greatly strengthen the manuscript and enhance its contribution to the literature.

**Do you want your identity to be public for this peer review?** For information about this choice, including consent withdrawal, please see our Privacy Policy

Reviewer #1: No

Reviewer #2: **Yes:**  Alejandro González Vidal

---

## [Author Response · Author response to Decision Letter 1]

26 Oct 2025

Dear Editor,

We are extremely grateful for the opportunity to revise our manuscript and to incorporate the insightful suggestions provided by the reviewers. Several aspects have been addressed, including a more detailed description of our methods and randomization, as well as an improved statistical description with additional data, such as effect sizes of the study findings. We also expanded the discussion of our study's limitations, highlighting potential biases and the need for caution in generalizing the results. Furthermore, we revised the text to better standardize terms such as mock OSCE and to avoid unnecessary repetitions, aiming for greater fluency and clarity in the reading.

Following a careful statistical review, we have updated the analysis of summative OSCE performance by adopting a two-tailed approach in place of the original one-tailed analysis. This revision better reflects standard statistical practice and provides a more conservative and robust assessment of the intervention effect. Consequently, the corresponding p-value, confidence intervals, and effect size estimates have been updated accordingly in the Results and Abstract sections of the revised manuscript.

One of the reviewers recommended a modification to the manuscript title, with which we concur. Could you please advise on the procedure for updating the title? The proposed new title is: "Empowering Medical Students: Peer-Led OSCE Reduces Anxiety and May Enhance Test Performance." Please note that we have already incorporated this revised title into the updated manuscript file.

All questions and suggestions raised by the reviewers are addressed in detail in the following sections of this document.

We are confident that the revised version is more robust and aligns more closely with the publication criteria of PLOS ONE. We remain available for any further clarifications you may consider necessary.

Kind regards,

Leonardo Mateus de Lima

Universidade de São Paulo – Faculdade de Medicina

Reviewers' comments:

Reviewer's Responses to Questions

Comments to the Author

1. Is the manuscript technically sound, and do the data support the conclusions?

Reviewer #1: Partly

Reviewer #2: Yes

2. Has the statistical analysis been performed appropriately and rigorously?

Reviewer #1: Yes

Reviewer #2: Yes

3. Have the authors made all data underlying the findings in their manuscript fully available?

Reviewer #1: Yes

Reviewer #2: Yes

4. Is the manuscript presented in an intelligible fashion and written in standard English?

Reviewer #1: Yes

Reviewer #2: Yes

5. Review Comments to the Author

Reviewer #1: Manuscript title: Empowering Medical Students: Peer-Led OSCE Reduces Anxiety and Enhances Test Performance

Major Comments:

1. randomization and sample composition: the manuscript would benefit from a more detailed account of the randomization process. It is currently unclear how the allocation sequence was generated or whether stratification was employed to balance the demographic or academic variables across groups. Additionally, the potential for selection bias should be acknowledged, particularly because students volunteered to participate in the mock OSCE. This self-selection could influence both anxiety levels and performance outcomes of the participants.

Thank you for your valuable comments and suggestions.

1.1. Randomization: We corrected the original version with more randomization details (lines 105 – 109 – revised version). We used simple randomization, without any stratification.

1.2. Selection bias: We acknowledge the potential for selection bias, as participation in the study, including the mock OSCE, was voluntary. However, all 5th-semester students volunteered to participate, and the final sample represents a substantial proportion of the cohort. Furthermore, randomization was performed only after the final list of volunteers was compiled, which helped ensure a balanced allocation between groups.

It is also important to note that, although seven students initially randomized to the intervention group were unable to participate in the mock OSCE, they were reassigned to the control group prior to the intervention, preserving the integrity of group comparisons. We have added a statement acknowledging this limitation in the manuscript. (lines 348 – 366, revised version).

2. power analysis and effect size: while a power analysis is mentioned, the underlying assumptions are not provided in the manuscript. The authors should specify the expected effect size, standard deviation, and dropout rate used for the calculation. This information is important for assessing whether the sample size was sufficient to detect meaningful differences between groups. Reporting effect sizes for the main comparisons further contextualizes the statistical findings.

2.1. Power analysis:

We have now clarified the assumptions used in the sample size calculation in the manuscript. Specifically, the calculation was based on state anxiety as the primary outcome, with a 10-point difference considered clinically meaningful. Data from a pilot study conducted with a previous cohort of 65 fifth-semester students showed a mean STAI score of 59.8 and a standard deviation of 12.83. Assuming a two-tailed test with α = 0.05, power = 0.80, and an estimated dropout rate of 10%, the required sample size was calculated to be 30 participants per group. These details have now been added to the Methods section (lines 167 – 174, revised version).

2.2. Effect size:

We have added effect size analysis, as follows:

For the comparison of means in anxiety state levels, we included Cohen’s d with the confidence interval (d = 0.692). (line 224).

For OSCE performance, the effect size measure used was Rosenthal’s r (r = 0.277). (line 244).

3. content overlap between assessments: there is a noted overlap in the content between the mock OSCE stations and the summative exam. This raises the possibility that performance improvements were influenced by familiarity rather than anxiety reduction. Although the authors briefly acknowledge this issue, a more explicit discussion of its potential impact on the results is required. The distinction between cognitive reinforcement and affective modulation must be clearly highlighted.

We acknowledge the reviewer’s point. Certainly, familiarity was an important factor in this study. Regarding content, although many stations covered similar themes across the examinations, the structure and required tasks were distinct. In the LGBTQIA+ health station, however, there was a considerable degree of similarity in what was being asked, which may have contributed to a content-familiarity effect. Familiarity is also related to understanding the structure of the examination itself—how to behave, how to manage time, and how to navigate the process—not only in terms of conceptual knowledge.

As for the reduction in anxiety as a possible contributing factor to improved performance, this remains a plausible explanation; however, the study did not objectively assess this association. In fact, our findings showed that, within our sample, there was no statistically significant correlation between anxiety levels and performance.

(lines 323 – 327).

4. interpretation of performance outcomes: the observed difference in OSCE scores, although statistically significant, was modest. The manuscript should address the clinical or educational relevance of these differences. Would this change have any implications for grading thresholds, remediation decisions or student confidence in clinical environments? Clarifying this would enhance the practical utility of these findings.

We agree that the difference in OSCE performance scores was modest, non-significant, and should be interpreted with caution. From an educational standpoint, this difference would likely not be sufficient to affect grading thresholds or remediation decisions. Nevertheless, we believe this finding may reflect a small, yet meaningful, benefit in student preparedness or confidence. We have now addressed this point in the Discussion section and acknowledged the need for future studies with larger samples and more robust designs to determine whether mock OSCE participation can produce clinically or educationally significant effects.

Minor Comments

5. The manuscript does not specify whether the assumptions of normality were tested before applying parametric tests. Given the small sample size, this needs to be confirmed. A brief note on how the data distribution was assessed would be enough.

In line 163, we mention that the Shapiro–Wilk and D’Agostino normality tests were applied, as appropriate, to each sample in order to verify the distribution and thereby select the correct parametric or non-parametric test. We will reinforce this information for greater clarity.

6. writing and redundancy: the text includes several repetitive elements, particularly in the Abstract and Introduction sections. The notion that OSCEs are anxiety-inducing was stated multiple times with minimal variation. Streamlining these sections will improve readability.

Repetition: we excluded the opening of the paragraph on OSCE and anxiety (discussion section) and other small sentences to avoid unnecessary redundancy.

7. visual representation of data: figures and tables are informative, but the presentation could be enhanced by including graphical summaries of anxiety scores and OSCE outcomes. Box plots or bar graphs with confidence intervals could facilitate reader interpretation.

We added graphs and tables to better illustrate our data.

Reviewer #2: This is a well-structured and timely study that addresses a relevant challenge in medical education. The idea of a peer-led OSCE is innovative and clearly has practical implications for reducing test anxiety and potentially improving performance. The manuscript is generally clear and the methodology is appropriate. However, I encourage the authors to expand their discussion of potential biases (e.g., self-selection, lack of blinding) and improve statistical reporting by including effect sizes and confidence intervals. Addressing these points will greatly strengthen the manuscript and enhance its contribution to the literature.

Dear Reviewer,

We are very grateful for your thoughtful comments and suggestions. Following the opportunity to revise our manuscript, we expanded the discussion regarding potential biases and limitations of our work and added details to improve the statistical reporting. We also revised some terms and reduced redundancies and repetitions to make the reading more concise and fluid.

6. PLOS authors have the option to publish the peer review history of their article (what does this mean?). If published, this will include your full peer review and any attached files.

Do you want your identity to be public for this peer review? For information about this choice, including consent withdrawal, please see our Privacy Policy.

Reviewer #1: No

Reviewer #2: Yes: Alejandro González Vidal

Reviewer Comments on the Manuscript

"Empowering Medical Students: Peer-Led OSCE Reduces Anxiety and Enhances Test

Performance" (PONE-D-25-10796)

General Assessment

This is a well-conceived and relevant study addressing the impact of peer-led mock OSCEs on anxiety and test performance among medical students. The topic is timely and contributes to the literature on cost-effective and student-centered assessment preparation strategies. The manuscript is clearly structured and presents promising results. However, several points should be addressed to strengthen the scientific rigor, clarity, and interpretability of the findings.

Major Points

Study Design and Potential Bias

The non-blinded design and the voluntary nature of the intervention group raise concerns about possible selection bias. Although this limitation is acknowledged, it deserves deeper consideration.

Recommendation: Provide more detail on how baseline characteristics were compared between groups and whether any pre-intervention anxiety or performance indicators were available. Clarify how the study design attempted to minimize self-selection effects.

We acknowledge the potential for selection bias, as participation in the study, including the mock OSCE, was voluntary. However, all 5th-semester students volunteered to participate, and the final sample represents a substantial proportion of the cohort. Furthermore, randomization was performed only after the final list of volunteers was compiled, which helped ensure a balanced allocation between groups.

It is also important to note that, although seven students initially randomized to the intervention group were unable to participate in the mock OSCE, they were reassigned to the control group prior to the intervention, preserving the integrity of group comparisons. We have added a statement acknowledging this limitation in the manuscript. (lines 354 – 360, revised version).

Generalizability

The study is based on a single institution with a relatively small and predominantly

female sample, which limits generalizability.

Recommendation: Clearly state these limitations in the discussion and suggest future multicenter or larger-scale studies to validate the findings.

We added the following passage at line 361 of the revised manuscript:

The generalizability of the study findings should be interpreted with caution. This was a single-center study with a relatively small sample size and a predominance of female participants (over 70%). As such, the results may not be fully representative of broader medical student populations with different demographic or institutional profiles. Future multi-center studies with more diverse and larger samples are needed to confirm these findings.

Statistical Reporting

While appropriate statistical tests were used, key measures such as effect sizes and confidence intervals are missing. These are essential for interpreting the magnitude and clinical relevance of the effects.

Recommendation: Include effect sizes (e.g., Cohen’s d) and 95% confidence intervals for primary and secondary outcomes.

We added effect size analysis, as follows:

For the comparison of means in anxiety state levels, we included Cohen’s d with the confidence interval (d = 0.692).

For OSCE performance, the effect size measure used was Rosenthal’s r (r = 0.277).

Qualitative Insights Missing

The intervention's success likely depends on psychological and educationa

---

## [Editor Report · Decision Letter 1]

21 Dec 2025

Empowering Medical Students: Peer-Led OSCE Reduces Anxiety and May Enhance Test Performance

PONE-D-25-10796R1

Dear Dr. Mateus de Lima,

We’re pleased to inform you that your manuscript has been judged scientifically suitable for publication and will be formally accepted for publication once it meets all outstanding technical requirements.

Kind regards,

Lorenzo Faggioni, M.D., Ph.D.

Academic Editor

PLOS One

---

## [Editor Report · Acceptance letter]

PONE-D-25-10796R1

PLOS One

Dear Dr. Mateus de Lima,

I'm pleased to inform you that your manuscript has been deemed suitable for publication in PLOS One. Congratulations! Your manuscript is now being handed over to our production team.

Kind regards,

on behalf of

Dr. Lorenzo Faggioni

Academic Editor

PLOS One